# Potential of organizing unmarried adolescent girls and young women into self-help groups for a better transition to adulthood: Findings from a cross-sectional study in India

**Jaleel Ahmad**[1]☉*, **Avishek Hazra**[1]☉, **Kumudha Aruldas**[2], **Arima Singh**[3],
**Niranjan Saggurti**[1]

1 Population Council, New Delhi, India, 2 Christian Medical College, Vellore, Tamil Nadu, India, 3 IPE Global Ltd, Jaipur, Rajasthan, India

☉ These authors contributed equally to this work.
* jahmad@popcouncil.org

**Data Availability Statement:** The data, study tools and a readme file explaining about the data are

## Abstract

It is essential to equip adolescents with the right information and appropriate skills for a quality transition to their adulthood. This study examines the individual agency of unmarried adolescent girls and young women (AGYW) who were organized into self-help groups (SHG) as compared to those who were not in groups. The paper uses data from a cross-sectional survey conducted with 872 unmarried AGYW aged 15–21 years from 80 villages across two districts of Uttar Pradesh, India. The dependent variables were AGYW's financial independence, collective action, decisionmaking, mobility, self-expression, generalized perceived self-efficacy, gender norms attitudes, and attitudes toward violence. The primary independent variables were group membership and the duration of the membership. Bivariate and multiple logistic regression analyses were conducted to examine the relationship between group membership and various components of individual agency. More than half of the respondents, with an average age of 18 years were enrolled in school or college and one-third had 12 or more years of education. The group members, compared to non-members, were significantly more likely to be financially independent (odds ratio [OR] = 2.29, p<0.01), to take collective action for entitlements (OR = 3.80, p<0.01), and to have progressive attitudes toward gender roles and norms (OR = 1.43, p<0.05). A longer duration of group membership increases the likelihood of financial independence, collective action, and decisionmaking ability. The study highlights the need for further investment in adolescent girls' programming and highlights the potential of organizing AGYW into SHG and using the 'platform' to bring change in their lives and consequential individual agency.

## Introduction

Globally, millions of adolescents are transitioning to adulthood every year and among them about half are females. The young population can be an asset to any nation. They can drive

available on the Harvard data verse (https://doi.org/10.7910/DVN/NQYBPM).

**Funding:** The Population Council was funded for this study by the Bill & Melinda Gates Foundation (Grant OPP1141832). The funder provided support in the form of salaries for authors [JA, AH, KA], but did not have any additional role in the study design, data collection and analysis, decision to publish, or preparation of the manuscript. The specific roles of these authors are articulated in the 'Author Contributions' section. At the time of conduct of the study and manuscript development, AS was employed by the organization that was working with YWSHGs. Currently, AS is employed by a commercial organization that did not have any role in this study.

**Competing interests:** The authors declare no individual or organizational competing interest exist. Commercial affiliation of one author (AS) does not alter our adherence to PLOS ONE policies on sharing data and materials.

productivity if they are provided a quality transition to adulthood, i.e. equipped with proper information and skills [1]. There is a growing recognition that achieving sustainable development goals (SDG) by 2030 requires a deliberate investment in adolescents, particularly girls [2–6]. Evidence shows that in poor settings adolescent girls often face psychosocial challenges such as mobility restrictions, lower self-efficacy, and lesser decisionmaking power, and hardly exercise their choice in personal matters [7]. They have limited opportunities for education, skill acquisition, and personal development [8]. Further, they must confront gender inequalities and gender role expectations [3, 5, 7, 9–13]. Adolescents often do not have a safe environment to socialize, and lack the opportunity to learn life skills or to get essential and correct information on health and other issues that concern their lives [5, 14, 15] and many are neither equipped nor supported to make a healthy transition to adulthood.

The vast majority of adolescents live in developing countries and more than half of the world's adolescents live in Asia with the highest number in India [16]. One-fourth of India's population consists of adolescents [11]. Uttar Pradesh (UP), the most populous state of India, contributes more than 40 million adolescents and young girls between 15–21 years of age to the country's total population [17]. The Government of India and the State government have launched several schemes and programs for adolescent girls to provide opportunities to improve their health and well-being [18]. Many services have not reached the target group adequately for many reasons including lack of (correct) information among adolescents and /or their parents about schemes, absence of proper guidance to access schemes, challenges in accessing the services due to poor implementation, and insufficient services from the system [9].

A non-governmental organization in rural UP has mobilized unmarried adolescent girls and young women (AGYW) in the age group of 13–21 years and organized them into Young Women Self-Help Groups (YWSHG) with the objective of making generational change through the girls' psychosocial development. The organization mainly targeted girls who had dropped out of school and were from families that already had at least one ever-married woman as a member of another self-help group (SHG) of ever-married women formed by the organization. A YWSHG consists of 10–12 AGYW as its members. The AGYW recieved trainings on basic financial literacy, menstrual health and hygiene, vocational training for income generation, leadership skills, rights, and accessing entitlements to social schemes and services. These trainings were meant to increase girls' decisionmaking power, self-efficacy, mobility, and collective action for their entitlements which are the key elements of empowerment. The YWSHG were also assumed to provide a safe space to AGYW where they could discuss their life challenges, learn life skills, and support each other in their needs.

There is growing evidence that suggests that developing girls' and women's skills, providing opportunities and access to resources, and offering vocational training can improve their health and social outcomes. Girl-centered programs in Bangladesh, Egypt, and Uganda demonstrated the transformations of girls' lives through vocational and life-skill training and engaging them to improve their confidence, self-efficacy, and mobility, by targeting gender norms [19–21]. The work with AGYW using the YWSHG is a unique program that included multiple activities mentioned in the previous paragraph with a focus on building the capacity to improve self-efficacy to mobilize AGYW to take collective action for their rights and entitlements. There is a lack of evidence on whether organizing AGYW into YWSHG at a large scale makes any difference in their empowerment status. This study aimed to explore whether organizing AGYW into YWSHG, exposed them to various training on life skills can make any difference in their ability to make decisions, financial independence, mobility, collective action, self-expression, self-efficacy to deal with difficult situations, attitude towards gender norms and access to resources in comparison with AGYW who were not members of YWSHG.

## Methods

### Ethics statement

The study participants were contacted and interviewed at their homes by trained female research assistants. Along with the oral informed consent of the girls, the guardian's written informed consent was obtained to interview girls below the age of 18 years. Further, the interviews were conducted maintaining the privacy and used the Computer Assisted Personal Interviewing technique. Semi-structured questionnaires, in the local language Hindi, which included various psychometric tools tested with age groups in Indian settings, were used in the survey. The study protocol and tools were reviewed and approved by the Institutional Review Board of Population Council, the research institution that conducted this study. The protocol approval number is 740 and was approved on May 18, 2016.

### Data

A cross-sectional survey was conducted in 2016 among unmarried AGYW aged 15–21 years who were members of YWSHG with a comparison group who were not members. Two districts of UP, Amethi, and Sultanpur, with the highest number of YWSHG, were selected purposively for the study. All the Gram Panchayats (GPs), the lowest administrative units, with three or more YWSHG were listed for the sampling purpose. Twenty GPs from each of these two districts were selected following systematic random sampling. The 40 GPs from these two districts had 127 YWSHG and three to four of AGYW from each of these YWSHG were randomly selected for the interview from the list of members provided by the leaders of each group. To select AGYW who were not members of YWSHG, an equal number of 40 GPs without the presence of YWSHG were randomly selected from the same two districts from where the GPs with YWSHG were selected. The YWSHG and non-YWSHG GPs were matched in terms of the GP-wise proportion of Scheduled Caste (SC)/Scheduled Tribe (ST) population to the total population. Historically, the SCs and STs have been socially and economically marginalized segments in Indian society and the proportion of such a population within a geographic location influences many developmental parameters. For identifying respondents from non-YWSHG households, all unmarried AGYW aged 15–21 years were listed from 100 households of the randomly selected segments of the selected GPs. The eligible respondents were selected randomly from the list obtained through the house listing. In total, 872 AGYW—471 YWSHG members and 401 non-members were interviewed.

A rigorous quality-assurance process was followed in data collection. The data-entry package used in the interview was developed in CSPro software version 6.3 and had built-in logic and checks to reduce non-sampling errors. Further, one quality-assurance person was assigned to each survey team of 4–5 members to assess the data quality through spot checks and back checks, and supportive supervision. The field supervisors sent data daily to the study coordinator who analyzed data on an ongoing basis and discussed any data-related queries with the field teams.

### Measures

The primary measure of the study includes the individual agency of AGYW. The individual agency has several connotations in the literature and is defined as per its use [22]. In this study, the individual agency has been defined as the ability of AGYW to make decisions and have financial independence, mobility, collective action, self-expression, belief in their ability to deal with difficult situations, egalitarian gender norms, and access to resources. Each of these indicator consist of several questions or variables. Composite indices were created to summarize information for each indicator and were used as dependent variables in multiple logistic regressions. Each variable

considered under a composite index was dichotomized and assigned a score of either 0 or 1. The mean score for an index was calculated and considered as a cut-off to divide the scores into low and high. Scores above the mean were categorized as high and the rest as low.

## Dependent variables

A description of each of the dependent variable used in this paper is presented here. The questions on financial independence, collective action, decisionmaking, mobility, self-expression, and attitudes towards violence were adopted from tools that explored these issues at scale and are available in the public domain [23]. Tested scales were used for measuring the variables on generalized perceived self-efficacy [24] and gender norm attitudes [25].

**Financial independence.** The girls were considered financially independent if they had saved money, had a saving goal, calculated the number of days to achieve their saving goal, had a bank account in their name, were operating their bank account on their own and were aware of basic banking services such as depositing money, loans and keeping valuables in the bank.

**Collective action.** Collective action was defined as action taken by AGYW in a group, or along with other girls, to demand their entitlements from health workers, Panchayat members, or bank officials in the past six months of the study.

**Decisionmaking.** The girls were considered empowered in decisionmaking if they perceived that they can make decisions entirely on their own in matters that concern their education, health care, taking a job, spending their earned money, and decisions about marriage.

**Mobility.** Freedom of movement or physical mobility was measured if girls were allowed to visit their neighborhood shop or market, community program within the village, or places outside the village or neighborhood including a friend or relative's house, a place of entertainment like a theatre, or any program outside the village and a health facility without any escort.

**Self-expression.** Self-expression was measured by the ability of girls to express their opinion to their elders about the issues that concern them, to confront a person if he or she says or does something wrong, to express their feelings to their parents, and to tell their parents about their marriage decisions and autonomy to choose their life partner.

**Generalized perceived self-efficacy.** Generalized perceived self-efficacy was measured through a generalized self-efficacy scale developed and adapted in India. The scale was designed to assess personal agency i.e. confidence in dealing with difficult situations and circumstances, such as being able to find means and ways to get what she wants, dealing efficiently with unexpected events, knowing how to handle unforeseen situations, and ability to usually find several solutions if confronted with a problem [24].

**Gender norms attitudes.** Gender roles and attitudes were assessed using the Gender Equitable Attitude and Gender Equitable Norm Scales developed by C-Change [25]. The key variables include girls' views on the importance of their education compared to boys, the decision on when to marry, the boy's participation in domestic work and care of children by husband, freedom to do things, and decision on how to spend money.

**Attitudes toward violence.** Attitude toward wife-beating was measured to gauge the extent to which girls thought wife-beating by the husband was justified if (a) he finds her unfaithful, (b) she goes out without telling him, (c) she disagrees with the opinion of her husband, (d) she makes a mistake, (e) she does not care of her children to his satisfaction, (f) she gives birth only to girls, and (g) she refuses to have sexual relation with him.

## Independent variables

The independent variables were YWSHG group membership and the duration of the membership. The duration of YWSHG membership was categorized into 1–5, 6–11, 12–23, and 24 or

more months to examine its association with individual agency. The non-members were assigned '0' for the duration of the membership. Socio-demographic characteristics of adolescents were obtained based on single questions asked in the survey. Age was categorized into 15–19 and 20 or more years; caste into SC/ST, Other Backward Classes, and General; education into four categories–up to class 5, class 6–8, class 9–11, and class 12 or more. The variables of SHG membership of family members, receipt of vocational training, and exposure to mass media (either reads a newspaper or listen to the radio or watches television) were dichotomized into "Yes" and "No". A proxy variable of the standard of living index (SLI) was computed to denote a household's economic condition and was used in the analysis. The SLI computation was based on an additive index using four variables available in the dataset–the main source of drinking water, the main source of lighting for the household, the main fuel used for cooking in the household, and the type of house.

## Statistical analyses

Descriptive statistics and multiple logistic regression models were used to examine the associations between AGYWs' agency indicators and YWSHG membership or duration of the membership. The socio-demographic characteristics of AGYW were taken as controlled variables in the regression models.

## Results

### Socio-demographic characteristics of adolescent girls and young women

The average age of the respondents was 18 years. The socio-demographic characteristics of AGYW were similar among the YWSHG members and non-members, except for two factors–education and family member's SHG membership (Table 1). More than half of the AGYWs (57 percent) were currently in school or college at the time of the interview and about one-third had completed class 12 or more. A significantly higher proportion of non-members (39 percent) than YWSHG members (35 percent) had completed class 12 or more (OR = 1.5, p<0.05). Among AGYW, who were currently in school, more than two-thirds wanted to continue their education till graduation, and among those who dropped out of school or college, about 40 percent wanted to be a graduate or even postgraduate. A higher proportion of YWSHG members were from SHG families (60 percent) compared to non-members (18 percent).

### Association between YWSHG membership and individual agency

Table 2 shows that a significantly higher proportion of YWSHG members as compared to non-members scored high for financial independence (46 percent vs 28 percent), collective action (26 percent vs 6 percent), decisionmaking (27 percent vs 21 percent), and mobility (26 percent vs 21 percent). Also, a significantly higher proportion of AGYW from YWSHG justified wife-beating by the husbands (52 percent) compared to non-members (44 percent). There was no statistical difference between the two groups of participants around gender norm attitudes, self-expression, and generalized self-efficacy.

The logistic regression analyses showed that even after controlling for respondent's age, education, school enrolment, vocational training, exposure to mass media, caste, SHG membership of family members, household's SLI, YWSHG members, as compared to non-members, were twice more likely to have financial independence (OR = 2.29, p<0.01), four times more likely to take collective action (OR = 3.8, p<0.01), one-and-half times more likely to have an egalitarian attitude toward gender roles and norms (OR = 1.43, p<0.05) (Table 2).

**Table 1. Percent distribution of AGYW by select socio-demographic characteristics.**

| Characteristics | Categories | Total | Non-YWSHG | YWSHG | Adjusted OR |
|---|---|---|---|---|---|
| | | N = 872 | N = 401 | N = 471 | (CI, 95%) |
| Age (in years) | 15-19[R] | 84.7 | 85.0 | 84.5 | – |
| | 20–25 | 15.3 | 15.0 | 15.5 | 0.96 (0.61–1.52) |
| Education (years of schooling) | Up to class 5[R] | 8.3 | 10.7 | 6.2 | – |
| | Class 6 to 8 | 21.3 | 17.5 | 24.6 | 1.03 (0.56–1.19) |
| | Class 9 to 11 | 33.8 | 33.2 | 34.4 | 2.27 (1.45–3.57)** |
| | Class 12 or more | 36.6 | 38.7 | 34.8 | 1.48 (1.01–2.17)* |
| Caste[a] | General[R] | 23.9 | 19.5 | 27.6 | – |
| | SC/ST | 31.7 | 38.4 | 25.9 | 0.60 (0.38–0.92) |
| | Other Backward Classes | 44.5 | 42.1 | 46.5 | 0.98 (0.66–1.45) |
| Standard of living index | Low[R] | 31.1 | 32.4 | 29.9 | - |
| | Middle | 53.0 | 54.6 | 51.6 | 0.92 (0.65–1.29) |
| | High | 15.9 | 13.0 | 18.5 | 1.45(0.89–2.37) |
| Currently in school/college | No[R] | 43.0 | 44.4 | 41.8 | – |
| | Yes | 57.0 | 55.6 | 58.2 | 0.87 (0.62–1.22) |
| From SHG family[a] | No[R] | 59.7 | 82.5 | 40.3 | – |
| | Yes | 40.3 | 17.5 | 59.7 | 7.0 (5.04–9.72)** |

**Note:**

a = chi-square test, $p < 0.001$; CI = confidence interval; R = Reference category; OR = odds ratio

*$p < 0.05$

**$p < 0.01$; In regression model group membership was taken as the dependent variable in the regression model.

# = The study aimed to interview girls of age 15–21 years, but the data contained information of 28 unmarried girls of age 22 to 25 years.

Findings further show that the duration of YWSHG membership of AGYW was significantly associated with higher financial independence, collective action, decisionmaking, and mobility ($p < 0.05$) (Table 3).

The logistic regression analysis indicated that AGYW who were members of YWSHG for 24 or more months, as compared to the non-members, were two and half times more likely to have financial independence (OR = 2.5, $p < 0.01$), five times more likely to take collective action for their entitlements (OR = 5.3, $p < 0.01$) and twice more likely to have decision ability on matters that concerns them (OR = 2.0, $p < 0.01$) (Table 4). Duration of YWSHG membership did not have any significant relationship with self-expression and an egalitarian attitude toward gender norms.

## Discussion

Adolescents and young women's membership in SHGs have a positive association with financial independence, collective action, and decisionmaking–the three critical individual level elements for the quality transition to adulthood. Further, AGYW with longer association with YWSHG had higher financial independence, were more likely to have decisionmaking power, and were more likely to participate in collective action. These could be the result of increased solidarity among the members with a longer duration of association in the group. Evidence suggests that adolescence is the formative age for developing basic financial-management skills like regular saving habits with saving goals which can help them to mitigate their future economic vulnerabilities as a consequence of poverty [1, 4]. Furthermore, economic empowerment and access to money are important for girls to mitigate financial emergencies, particularly for education and medical emergencies [4]. The SHGs are generally based on the

**Table 2. Association between AGYWs' individual agency related indicators and YWSHG membership.**

| | NON-YWSHG (N = 401) % | YWSHG (N = 471) % | Adjusted OR (95% CI) |
|---|---|---|---|
| Financial independence (High)[a] | 28.2 | 46.3 | 2.29 (1.62–3.25)** |
| Collective action (High)[a] | 6.2 | 25.7 | 3.82 (2.32–6.31)** |
| Decisionmaking (High)[a] | 20.9 | 27.2 | 1.34 (0.94–1.93) |
| Mobility (High)[a] | 20.7 | 25.7 | 1.09 (0.75–1.60) |
| Self-expression (High) | 45.4 | 48.4 | 1.01 (0.74–1.36) |
| Generalized self-efficacy (High) | 50.9 | 45.0 | 0.72 (0.53–0.98)* |
| Attitude toward gender norms—Egalitarian | 45.4 | 48.0 | 1.43 (1.04–1.95)* |
| Attitude toward domestic violence (justified)[a] | 43.9 | 52.0 | 1.63 (1.19–2.22)** |

**Note**: CI = confidence interval; OR = odds ratio; Each of the dependent variables–financial independence, collective action, decisionmaking, mobility, self-expression, and generalized self-efficacy were dichotomized as low (0) and high (1), attitude toward gender norms as non-egalitarian (0) and egalitarian (1), and attitude toward gender norms as not justified (0) and justified (1). The key independent variable is YWSHG membership (non-member is the reference category). Each of the regression models was adjusted for AGYW's age, education, school enrolment, vocational training, exposure to mass media, caste, SHG membership of family members, and household's standard of living index.

a. z-test, p<0.05

*p<0.05

**p<0.01

micro-finance model that encourages savings and builds collective efficacy to address the marginalization of SHG members. The findings of this study indicate the potential of YWSHG that allow space for AGYW to take collective action to avail themselves of entitlements such as demand for services from health workers or local administration or bank officials. Such capacities reduce their dependency on others to achieve a better life in the future. Interventions to build the capacity of girls in Uganda show similar results [21]. Empowering girls with skills and knowledge can improve their life outcomes [26]. Given this, YWSHG can play a critical role in building adolescents' capacities, skills, and choices in their lives.

Although financial independence and the ability to take collective action was higher among the members of YWSHG, but membership did not make much difference on other aspects such as mobility, self-expression, and attitude towards gender norms. Restricted mobility and limited agency among unmarried AGYW were also reported in the "Youth in India" study

**Table 3. Percentage distribution of AGYW who scored high in individual agency indicators by the duration of YWSHG membership.**

| | | Duration of membership in months | | | |
|---|---|---|---|---|---|
| | Not member (N = 401) | 1–5 (N = 64) | 6–11 (N = 89) | 12–23 (N = 229) | 24 or more (N = 89) |
| Financial independence (High)[a], % | 28.2 | 34.4 | 44.9 | 47.6 | 52.8 |
| Collective action (High)[a], % | 6.2 | 31.3 | 21.3 | 22.7 | 33.7 |
| Decisionmaking (High)[a], % | 20.9 | 21.9 | 23.6 | 26.2 | 37.1 |
| Mobility (High)[a], % | 20.7 | 15.6 | 16.9 | 30.6 | 29.2 |
| Self-expression (High), % | 45.4 | 42.2 | 51.7 | 48 | 50.6 |
| Generalized self-efficacy (High), % | 50.9 | 54.7 | 43.8 | 44.1 | 41.6 |
| Attitude toward gender norms–Egalitarian (High), % | 45.4 | 53.1 | 52.8 | 45.4 | 46.1 |
| Attitude toward domestic violence (justified), % | 43.9 | 54.7 | 53.9 | 49.8 | 53.9 |

**Note:**

[a]Chi-square test, p<0.05

**Table 4. Influence of duration of YWSHG membership on AGYW individual agency related indicators.**

| | Duration of membership in months | | | |
| | Odds ratios (95% CI) | | | |
| | 1–5 months | 6–11 months | 12–23 months | 24 or more months |
|---|---|---|---|---|
| Financial independence (High) | 1.68 (0.89–3.17) | 2.08 (1.21–3.56)** | 2.49 (1.65–3.74)** | 2.51 (1.47–4.23)** |
| Collective action (High) | 5.09 (2.45–10.56)** | 3.23 (1.61–6.47)** | 3.16 (1.80–5.53)** | 5.34 (2.80–10.18)** |
| Decisionmaking (High) | 1.08 (0.55–2.13) | 1.18 (0.66–2.09) | 1.23 (0.81–1.89) | 2.01 (1.18–3.41)** |
| Mobility (High) | 0.73 (0.33–1.60) | 0.61 (0.31–1.19) | 1.43 (0.92–2.22) | 1.12 (0.63–1.99) |
| Self-expression (High) | 0.79 (0.45–1.39) | 1.16 (0.72–1.88) | 0.98 (0.68–1.40) | 1.10 (0.68–1.79) |
| Generalized self-efficacy (High) | 0.79 (0.45–1.39) | 1.16 (0.72–1.88) | 0.98 (0.68–1.40)* | 1.10 (0.68–1.79)* |
| Attitude toward gender norms–(Egalitarian) | 1.62 (0.91–2.90) | 1.60 (0.97–2.64) | 1.33 (0.92–1.94) | 1.38 (0.83–2.29) |
| Attitude toward domestic violence (Justified) | 1.65 (0.94–2.93) | 1.65 (1.01–2.69)* | 1.51 (1.05–2.19)* | 1.10 (1.15–3.14)* |

**Note:** Each of the dependent variables–financial independence, collective action, decisionmaking, mobility, self-expression, and generalized self-efficacy was dichotomized as low (0) and high (1), attitude towards gender norms as non-egalitarian (0) and egalitarian (1), and attitude towards gender norms as not justified (0) and justified (1). The key independent variable is YWSHG membership duration (non-member is the reference category). Each of the regression models was adjusted for AGYW's age, education, school enrolment, vocational training, exposure to mass media, caste, SHG membership of family members, and household's standard of living index.

*$p < 0.05$

**$p < 0.01$

that was conducted in six states of India to assess the situation and needs of the adolescent and young population [12].

The study shows that YWSHG membership as well as the duration of membership did not affect attitude towards domestic violence. A significantly higher proportion of AGYW justified wife-beating. Further investigation is required to explore the reasons but the validation of the Gender Equitable Men (GEM) scale by Lung Vu showed that 70 percent girls of aged 10–24 years reported substantial support for inequitable gender norms, and women expected to tolerate violence to keep their family together [27]. As gender norms are followed generationally and reinforced by the family and community, it is harder to change adolescents' mindset through interventions of short duration.

YWSHG provides a platform for adolescent girls and young women to come together and voice their opinions. This strategy can also be used to make girls aware of health issues such as anemia which is a major health concern among girls in their second decade of life. A randomized control trial in Uganda showed that "combined" interventions have more effectiveness than a "single" intervention for the empowerment of adolescents to health behaviors [19].

Most of the YWSHG members are girls who drop out of school and uniting them in a platform can be a good strategy to engage them in life skill activities that can delay their chances of early marriage and childbearing. Studies have demonstrated that engaging young girls through a multiple economic empowerment approach combined with health education and skill development at the community level helps in delaying the age of marriage and improving health and social outcomes [20, 28].

The study had certain limitations. First, the study findings are based on a cross-sectional survey that cautions against establishing any cause-effect relationship. Longitudinal studies are better suited to examine the ways YWSHG membership changes the lives of adolescents. However, the results from the adjusted regression analysis showed a direction in the relationship between YWSHG membership and AGYWs' agency. Second, the study was done in a geographic location that had a high concentration of YWSHG. This might have had an indirect influence on girls' empowerment as the higher number of YWSHG might have a better chance

of taking collective action. Third, the selection bias of AGYW to be YWSHG members cannot be ruled out. AGYW, whose mothers were SHG members, may have higher levels of empowerment and would therefore be more likely to join groups. The analysis controlled for this potential selection bias by including AGYWs' socio-demographic characteristics as well as SHG membership of family members as confounding factors in the multivariate regression models.

## Conclusion

Adolescent girls and young women are the critical segments of the population on which future development depends. The Women Deliver Conference in 2013 made a strong call to invest in adolescents and young people, particularly girls, to improve their health and wellbeing, skills, and capacities to address various social and economic problems of the current and future generations [29]. The Women Deliver Conference in 2016 reiterated that when an investment is made in girls and women, everybody wins [30]. Investment in adolescents and youth can open a window of opportunity for a demographic dividend [26]. To achieve sustainable development goals, adolescents and young women require the right information, skills, and access to services.

The study noted that AGYW aspired for higher education–both those who were currently in school or college and those who had dropped out of school. Education is critical for the overall development of an individual. The schools and colleges provide space and opportunities for AGYW to achieve their educational aspirations. Therefore, programs addressing AGYW lives must address the issues around drop out of girls from school and colleges. This study found that YWSHG members compared to non-members had greater individual agency, thus demonstrating that YWSHGs are safe spaces for AGYW in rural settings to channel their potential and energies. Such groups provide them space to interact and a safety net for taking collective action. The finding that AGYW members were twice more likely to make decisions on matters that concern them indicates that YWSHG help in transform gender attitudes and norms which are critical for girls' future lives. The findings thus demonstrate that YWSHG have the potential to empower young women to be agents to change and transform future generations. Therefore, it is important to continue the mobilization of AGYW particularly in rural settings in India. Under the National Rural Livelihood Mission, the Government of India is making a deliberate attempt to improve the livelihood opportunities and empower women through SHGs, it may be equally important to invest and support in developing and designing or replicating such models of organizing AGYW into YWSHG, to implement interventions that improve the skills and lives of adolescents and young women.

## Acknowledgments

We are extremely grateful to all the adolescent girls and young women who participated in this study, their parents and guardians for their consent to allow their daughters to be part of this study, the organization that mobilized and united these girls under YWSHG, their field staff for extending support during data collection, and the research investigators who conducted interviews. Special thanks to Diana Nolan for copyediting the article.

## Author Contributions

**Conceptualization:** Jaleel Ahmad, Avishek Hazra, Kumudha Aruldas.

**Formal analysis:** Jaleel Ahmad, Avishek Hazra.

**Investigation:** Jaleel Ahmad, Avishek Hazra, Arima Singh.

**Methodology:** Avishek Hazra.

**Project administration:** Jaleel Ahmad, Avishek Hazra.

**Software:** Jaleel Ahmad.

**Supervision:** Jaleel Ahmad, Avishek Hazra, Niranjan Saggurti.

**Visualization:** Niranjan Saggurti.

**Writing – original draft:** Jaleel Ahmad.

**Writing – review & editing:** Kumudha Aruldas, Niranjan Saggurti.

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
