## [Decision Letter · Decision Letter 0]

4 Nov 2020

PONE-D-20-29446

Potential of organizing adolescent girls and young women into self-help groups for an inter-generational change: Learnings from a cross-sectional study

PLOS ONE

Dear Dr. Ahmad,

Thank you for submitting your manuscript to PLOS ONE. After careful consideration, we feel that it has merit but does not fully meet PLOS ONE’s publication criteria as it currently stands. Therefore, we invite you to submit a revised version of the manuscript that addresses the points raised during the review process.

The academic Editor served as the second reviewer on this manuscript. The editor agreed with the first reviewer. The authors also need to add a clear set of actionable recommendations linked to the key findings. The authors are strongly encouraged to specifically address the comments from the reviews if they wish to revise and resubmit, to be seriously considered for publication.

We look forward to receiving your revised manuscript.

Kind regards,

Joseph Telfair, DrPH, MSW, MPH

Academic Editor

PLOS ONE

Additional Editor Comments:

The academic Editor served as the second reviewer on this manuscript. The editor agreed with the first reviewer. The authors also need to add a clear set of actionable recommendations linked to the key findings.

Journal Requirements:

"The authors have declared that no competing interests exist"

We note that one or more of the authors are employed by a commercial company: name of commercial company.

2.1. Please provide an amended Funding Statement declaring this commercial affiliation, as well as a statement regarding the Role of Funders in your study. If the funding organization did not play a role in the study design, data collection and analysis, decision to publish, or preparation of the manuscript and only provided financial support in the form of authors' salaries and/or research materials, please review your statements relating to the author contributions, and ensure you have specifically and accurately indicated the role(s) that these authors had in your study. You can update author roles in the Author Contributions section of the online submission form.

2.2. Please also provide an updated Competing Interests Statement declaring this commercial affiliation along with any other relevant declarations relating to employment, consultancy, patents, products in development, or marketed products, etc.  

Reviewers' comments:

Reviewer's Responses to Questions

**Comments to the Author**

1. Is the manuscript technically sound, and do the data support the conclusions?

Reviewer #1: Yes

2. Has the statistical analysis been performed appropriately and rigorously? 

Reviewer #1: I Don't Know

3. Have the authors made all data underlying the findings in their manuscript fully available?

Reviewer #1: No

4. Is the manuscript presented in an intelligible fashion and written in standard English?

Reviewer #1: No

5. Review Comments to the Author

Reviewer #1: - The title is vague since the focus is not evident until you read the abstract that the authors assess transition to adulthood. I would strongly recommend adding transition to adulthood to title to let focus be known.

-Copyediting and proofreading the entire manuscript is strongly recommended

-Authors state that “The Government of India and the State government have launched several schemes and programs for adolescents girls to provide opportunities to improve their health and well-being [19]. But the services have not reached the target group adequately for many reasons including insufficient awareness and lack of knowledge among adolescents.” Not sure how insufficient awareness and lack of knowledge among adolescents are different reasons. What are the other “many reasons?”

- Need to explicitly state measures and scales used for other researchers to replicate. Would strongly advise to add data collection instruments to appendix to understand context of scales and measures to see why variables were measured the way they were (e.g. standard of living, exposure to mass media, etc.).

- Information regarding the caste system would be helpful to provide context for readers.

- Would suggest in the discussion when authors state “These findings were similar to the findings of the youth survey [12],” authors should give context about the youth survey referenced since it currently reads as a youth survey that was part of the authors' study even though that is not the case.

- Great job to the authors for combining multiple programmatic approaches to promote individual agency to incorporate nutrition, menstrual health and hygiene.

6. PLOS authors have the option to publish the peer review history of their article (what does this mean?). If published, this will include your full peer review and any attached files.

Reviewer #1: No

---

## [Author Response · Author response to Decision Letter 0]

16 Jan 2021

Editor’s comments 

Comment: The academic Editor served as the second reviewer on this manuscript. The editor agreed with the first reviewer. The authors also need to add a clear set of actionable recommendations linked to the key findings.

Response: We have reviewed the manuscript and added a set of actionable recommendations in the last para of the discussion (line number 347-365) as;

“The study noted that AGWYs aspired for higher education – both those who were currently in school or college and those who had dropped out of school. Education is critical for the overall development of an individual. The schools and colleges provide space and opportunities for AGYWs to achieve their educational aspirations. Therefore, programs addressing AGYWs lives must address the issues around drop out of girls from school and colleges. This study found that YWSHG members compared to non-members had greater individual agency, thus demonstrating that YWSHGs are safe spaces for AGYWs in rural settings to channel their potential and energies. Such groups provide them space to interact and a safety net for taking collective action. The finding that AGYWs members were twice more likely to make decisions on matters that concern them indicate that YWSHGs help in transform gender attitudes and norms which are critical for girls’ future lives. The findings thus demonstrate that YWSHGs have the potential to empower young women to be agents to change and transform future generations. Therefore, it is important to continue the mobilization of AGYWs particularly in rural settings in India. Now that under the National Rural Livelihood Mission, the Government of India is making a deliberate attempt to improve the livelihood opportunities and empower women through SHGs, it may be equally important to invest and support in developing and designing or replicating such models of organizing AGYWs into YWSHGs, to implement interventions that improve the skills and lives of adolescents and young women.”

Comment 1: Please ensure that your manuscript meets PLOS ONE's style requirements, including those for file naming. The PLOS ONE style templates can be found at

Response: We have thoroughly reviewed the manuscript and ensured formatting according to PLOS ONE’s style requirements 

Comment 2: Thank you for stating the following in the Competing Interests section:

"The authors have declared that no competing interests exist"

We note that one or more of the authors are employed by a commercial company: name of commercial company.

Response: One of the authors (Arima Singh) was involved with the study and manuscript writing while she was associated with the organization that was working with YWSHGs. Currently, she is employed by IPE Global Pvt. Ltd., an international development consulting company that did not have any role either in conduct of the study or writing of the manuscript.

We have revised the ‘funding statement’ and ‘competing interests’ statement’ to reflect this (please see response in point number 2.1 and 2.2).

Comment 2.1: Please provide an amended Funding Statement declaring this commercial affiliation, as well as a statement regarding the Role of Funders in your study. If the funding organization did not play a role in the study design, data collection and analysis, decision to publish, or preparation of the manuscript and only provided financial support in the form of authors' salaries and/or research materials, please review your statements relating to the author contributions, and ensure you have specifically and accurately indicated the role(s) that these authors had in your study. You can update author roles in the Author Contributions section of the online submission form.

Response: We have revised the funding statement and included 

“The Population Council was funded for this study by the Bill & Melinda Gates Foundation (Grant OPP1141832). The funder provided support in the form of salaries for authors [JA, AH, KA], but did not have any additional role in the study design, data collection and analysis, decision to publish, or preparation of the manuscript. The specific roles of these authors are articulated in the ‘author contributions’ section. At the time of conduct of the study and manuscript development, AS was employed by the organization that was working with YWSHGs. Currently, AS is employed by a commercial organization that did not have any role in this study.”

Comment 2.2: Please also provide an updated Competing Interests Statement declaring this commercial affiliation along with any other relevant declarations relating to employment, consultancy, patents, products in development, or marketed products, etc. 

Response: Thank you for raising the concern. We carefully reviewed the details on competing interests through the link you provided (http://journals.plos.org/plosone/s/competing-interests). We have revised the Competing Interests Statement as below:

“The authors declare no individual or organizational competing interest exist. Commercial affiliation of one author (AS) does not alter our adherence to PLOS ONE policies on sharing data and materials.”

Based on your advice, we have included both an updated Funding Statement and Competing Interests Statement in the revised cover letter. 

Comment 3: We note that you have stated that you will provide repository information for your data at acceptance. Should your manuscript be accepted for publication, we will hold it until you provide the relevant accession numbers or DOIs necessary to access your data. If you wish to make changes to your Data Availability statement, please describe these changes in your cover letter and we will update your Data Availability statement to reflect the information you provide. 

Response: Please find below the data repository information (DOIs) to access the data and tools.

https://doi.org/10.7910/DVN/NQYBPM

Reviewer’s Comments 

Comment 1: Is the manuscript technically sound, and do the data support the conclusions?

Reviewer #1: Yes 

Response: Thank you for your appreciation

Comment 2: Has the statistical analysis been performed appropriately and rigorously? 

Reviewer #1: I Don't Know 

Response: We have applied appropriate statistical techniques to examine the study objectives.

Comment 3: Have the authors made all data underlying the findings in their manuscript fully available?

Reviewer #1: No 

Response: Please find below the data repository information (DOIs) to access the data and tools.

https://doi.org/10.7910/DVN/NQYBPM

Comment 4: Is the manuscript presented in an intelligible fashion and written in standard English?

Reviewer #1: No 

Response: We have revised the manuscript and done copyediting to avoid any typographical or grammatical errors. 

Comment 5: Review Comments to the Author 

5.1 Reviewer #1: - The title is vague since the focus is not evident until you read the abstract that the authors assess transition to adulthood. I would strongly recommend adding transition to adulthood to title to let focus be known. 

Response: We have reassessed the title. In light of the outcome measures we have used in the paper, we have revised the title of the manuscript as “Potential of organizing unmarried adolescent girls and young women into self-help groups for a better transition to adulthood: Findings from a cross-sectional study in India.

5.2 -Copyediting and proofreading the entire manuscript is strongly recommended 

Response: We have thoroughly reviewed the manuscript and did the copyediting to avoid spelling and grammatical errors.

5.3 -Authors state that “The Government of India and the State government have launched several schemes and programs for adolescents girls to provide opportunities to improve their health and well-being [19]. But the services have not reached the target group adequately for many reasons including insufficient awareness and lack of knowledge among adolescents.” Not sure how insufficient awareness and lack of knowledge among adolescents are different reasons. What are the other “many reasons?”

Response: We have revised the statement and expanded “many reasons”. The revised text in the manuscript (line number 61-67) is as follows:

“The Government of India and the State government have launched several schemes and programs for adolescents’ girls to provide opportunities to improve their health and well-being [18]. But many services have not reached the target group adequately for many reasons including lack of (correct) information among adolescents and /or their parents about schemes, absence of proper guidance to access schemes, challenges in accessing the services due to poor implementation and insufficient services from system [9]”.

5.4 - Need to explicitly state measures and scales used for other researchers to replicate. Would strongly advise to add data collection instruments to appendix to understand context of scales and measures to see why variables were measured the way they were (e.g. standard of living, exposure to mass media, etc.).

Response: We have elaborated the measures about the scales used in the paper and given proper citation to refer to the original sources from where we adopted the questions/scales. The instrument used for the present study is available at 

https://doi.org/10.7910/DVN/NQYBPM

We have explained the construction of standard of living index (Line no. 198-202) and exposure to mass media (Line no.197). The revised text is as below

SLI: “A proxy variable of standard of living index (SLI) was computed to denote a household’s economic condition and was used in the analysis. The SLI computation was based on an additive index using four variables available in the dataset – main source of drinking water, main source of lighting for the household, main fuel used for cooking in the household and type of house.”

Mass media exposure: “either reads newspaper or listen to radio or watches television”.

5.5 - Information regarding the caste system would be helpful to provide context for readers.

Response: We have added information on caste system. The text is added in Line no. 111-113 appears as: 

“Historically, the SCs and STs have been socially and economically marginalized segments in Indian society and the proportion of such a population within a geographic location influences many developmental parameters.”

5.6 - Would suggest in the discussion when authors state “These findings were similar to the findings of the youth survey [12],” authors should give context about the youth survey referenced since it currently reads as a youth survey that was part of the authors' study even though that is not the case.

Response: The context has been added to the reference of the youth survey. The revised text in Line no. 300-303 appears as:

 “Restricted mobility, and limited agency among unmarried AGYW was also reported in “Youth in India” study that was conducted in six states of India to assess the situation and needs of the adolescent and young population [12]”.

5.7 - Great job to the authors for combining multiple programmatic approaches to promote individual agency to incorporate nutrition, menstrual health and hygiene.

Response: Thank you for the appreciation.

---

## [Decision Letter · Decision Letter 1]

4 Feb 2021

PONE-D-20-29446R1

Potential of organizing unmarried adolescent girls and young women into self-help groups for a better transition to adulthood: Findings from a cross-sectional study in India

PLOS ONE

Dear Dr. Ahmad,

Thank you for submitting your manuscript to PLOS ONE. After careful consideration, we feel that it has merit but does not fully meet PLOS ONE’s publication criteria as it currently stands. Therefore, we invite you to submit a revised version of the manuscript that addresses the points raised during the review process.

In light of the challenge is finding competent reviewers, the editor served as the second reviewer. There remains concerns about editing (inconsistent places) that need to be addressed for any last publication considerations. See below for what needs to be specifically addressed. 

 Strongly recommend another round of copyediting (e.g. AGYW is mentioned in some parts of the paper while AGWY is used in other parts of the paper). The link to the data repository led to data files that can be downloaded. However, data is unable to be downloaded because the data use form requires identifiable information.

We look forward to receiving your revised manuscript.

Kind regards,

Joseph Telfair, DrPH, MSW, MPH

Academic Editor

PLOS ONE

Reviewers' comments:

Reviewer's Responses to Questions

**Comments to the Author**

1. If the authors have adequately addressed your comments raised in a previous round of review and you feel that this manuscript is now acceptable for publication, you may indicate that here to bypass the “Comments to the Author” section, enter your conflict of interest statement in the “Confidential to Editor” section, and submit your "Accept" recommendation.

Reviewer #1: All comments have been addressed

2. Is the manuscript technically sound, and do the data support the conclusions?

Reviewer #1: Yes

3. Has the statistical analysis been performed appropriately and rigorously? 

Reviewer #1: Yes

4. Have the authors made all data underlying the findings in their manuscript fully available?

Reviewer #1: Yes

5. Is the manuscript presented in an intelligible fashion and written in standard English?

Reviewer #1: Yes

6. Review Comments to the Author

Reviewer #1: The authors have addressed the reviewers' comments in the previous round of review. Strongly recommend another round of copyediting (e.g. AGYW is mentioned in some parts of the paper while AGWY is used in other parts of the paper). The link to the data repository led to data files that can be downloaded. However, data is unable to be downloaded because the data use form requires identifiable information.

7. PLOS authors have the option to publish the peer review history of their article (what does this mean?). If published, this will include your full peer review and any attached files.

Reviewer #1: No

---

## [Author Response · Author response to Decision Letter 1]

1 Mar 2021

Comment: Strongly recommend another round of copyediting (e.g. AGYW is mentioned in some parts of the paper while AGWY is used in other parts of the paper). The link to the data repository led to data files that can be downloaded. However, data is unable to be downloaded because the data use form requires identifiable information.

Response: We have thoroughly reviewed the manuscript including copyediting. We have removed the restriction from the data repository but as an institution policy only few identifiable information such as name, email ID and institution are required to download the data. The data can be accessed at https://doi.org/10.7910/DVN/NQYBPM.

---

## [Editor Report · Decision Letter 2]

4 Mar 2021

Potential of organizing unmarried adolescent girls and young women into self-help groups for a better transition to adulthood: Findings from a cross-sectional study in India

PONE-D-20-29446R2

Dear Dr. Ahmad,

We’re pleased to inform you that your manuscript has been judged scientifically suitable for publication and will be formally accepted for publication once it meets all outstanding technical requirements.

Kind regards,

Joseph Telfair, DrPH, MSW, MPH

Academic Editor

PLOS ONE
---

## [Editor Report · Acceptance letter]

8 Mar 2021

PONE-D-20-29446R2 

Potential of organizing unmarried adolescent girls and young women into self-help groups for a better transition to adulthood: Findings from a cross-sectional study in India. 

Dear Dr. Ahmad:

I'm pleased to inform you that your manuscript has been deemed suitable for publication in PLOS ONE. Congratulations! Your manuscript is now with our production department. 

Kind regards, 

on behalf of

Dr. Joseph Telfair 

Academic Editor

PLOS ONE